# Current Limitations of Automated Reflection Consolidation in LLMs for Clinical Note Extraction

Leong Ting Lui[1,2], Trista Hung[3], Fengyuan Che[4], Hao Wang[4], Bonaventure Ip[3], Rosa H. M. Chan[1,2]

*Abstract*—Large language models (LLMs) are increasingly used to extract clinical information from unstructured text, but systematic methods for optimizing their performance in specialized medical domains are not well established. Reflection, a prompt-based approach to improve performance, has shown promise in various settings but is usually applied to single instances rather than generalized solutions. This study explores whether open-source LLMs can consolidate multiple successful reflections into reusable prompt components for clinical text extraction tasks. We tested six LLMs on clinical notes describing endovascular thrombectomy procedures, each model extracting seven key variables. For incorrect outputs, up to five rounds of self-reflection were triggered, and three strategies were compared for guiding these reflections. Corrective reflections were then consolidated into generalized prompt components. While all models improved on individual instances following reflection, consolidation of reflections led to mixed results: some models showed modest overall improvement, while others did not. The process of summarizing effective reflections often resulted in the loss of essential details, limiting the benefits of prompt consolidation. These findings suggest that while reflection aids self-correction, effective autonomous generalization remains challenging and calls for structured human-in-the-loop oversight during the consolidation phase to preserve critical clinical information during prompt optimization.

*Index Terms*—Stroke Surgery; Large language models; Human-in-the-loop; Information extraction; Prompt engineering

## I. INTRODUCTION

The field of natural language processing has experienced transformative advancements with the advent of LLMs. These models exhibit remarkable capabilities in generating human-like text, comprehending complex medical instructions, and performing diverse healthcare tasks. Nevertheless, extracting structured, clinically relevant information from unstructured medical text remains challenging, especially in high-stakes applications where precision and reliability are paramount. LLMs have been extensively investigated for various medical tasks, including information extraction [1]–[3], discharge summary generation [4], patient-trial matching [5], and clinical text augmentation [6]. Researchers have pursued fine-tuning and prompt optimization techniques to enhance LLMs' performance in specialized medical tasks [7]–[10].

Recent studies demonstrate that inference-time compute can significantly enhance LLMs' performance, often surpassing the gains achieved by scaling model size alone [11]–[13]. Methods such as chain-of-thought prompting [14], episodic memory-based reflection [15], and self-consistency [16]–[18] have proven effective for reasoning-intensive tasks. Reflective prompting [19], where LLMs identify and correct their own errors, has similarly improved clinical question answering in the medical domain [20]–[22].

Clinical information extraction requires adaptability to variations in documentation styles, medical practices, and regional standards [23], [24]. For instance, anticoagulant prescriptions may differ by region or institution, and definitions such as obesity also vary (e.g. BMI $>30$ in the West, $>25$ in East Asia). Such variability makes this an ideal use case for reflection-based methods, which can adapt prompts dynamically at inference time. However, most studies to date have focused on improving individual instances rather than enabling LLMs to systematically learn from past errors to enhance performance across similar cases.

One reason could be that current prompt optimization techniques are often evaluated on broad clinical benchmarks such as MedQA [25], MedMCQA [26], and PubMedQA [27]. While these datasets provide medical questions covering broad topics [28], their diversity can obscure domain-specific complexities, making it difficult to distill reusable insights across cases that may be only loosely related. In contrast, we employ a homogeneous dataset derived from clinical notes on endovascular thrombectomy procedures. Instead of viewing self-reflection as merely a tool for correcting individual errors, we examine the LLMs' capacity to generalize insights from prior reflections and apply them to new but contextually similar cases. This approach may be especially valuable in scenarios where high-quality annotated data is scarce and insufficient for fine-tuning. Homogeneity is key to testing the hypothesis that error patterns will be recurrent and thus amenable to consolidated correction strategies.

The contributions of this work extend beyond performance evaluation to provide fundamental insights into LLM reasoning capabilities. First, we present a reflection-based framework for iterative prompt refinement using LLMs, grounded in the hypothesis that insights from earlier reflections can be consolidated to guide future predictions. Second, we systematically evaluate the effectiveness of this approach using a

Corresponding author: rosachan@cityu.edu.hk

[1] Hong Kong Centre for Cerebro-cardiovascular Health Engineering, Hong Kong SAR

[2] Department of Electrical Engineering, The City University of Hong Kong, Hong Kong SAR

[3] Division of Neurology, Department of Medicine and Therapeutics, The Chinese University of Hong Kong, Hong Kong SAR

[4] Department of Neurology, Linyi People's Hospital, China

This work was supported fully by InnoHK Project at Hong Kong Centre for Cerebro-cardiovascular Health Engineering (COCHE).

homogeneous dataset. While our framework shows that LLMs can self-correct through reflection on individual cases, our findings also reveal key limitations in the LLM's autonomous ability to generalize such reflections. To address this challenge, we advocate for a human-in-the-loop refinement process during the consolidation phase, where clinical expertise can help preserve essential insights and ensure the reliability of reflection-based prompt optimization.

## II. METHOD

### A. Dataset

This dataset consists of 115 real-world clinical notes documenting endovascular thrombectomy procedures for acute ischemic stroke [29]. Each note captures patient presentations, preoperative assessments (including NIHSS scores and imaging findings), procedural steps (such as sedation protocols, vascular access, device selection, microcatheter guidance, stent retriever deployment, and aspiration technique). These notes also document intraoperative clinical decisions (e.g., switching catheters when encountering high-resistance thrombi) and acute complications or findings (e.g., minimal hemorrhage, reperfusion injury). Detailed timelines are provided for each stage, including insertion of guiding catheters, microcatheters, stent deployment, clot retrieval, and final revascularization outcomes (mTICI scores). Furthermore, the notes include post-operative management plans, such as blood pressure control, repeat head CT scans, and instructions for possible anticoagulation therapy, offering insight into routine clinical practice. The homogeneity of these notes provides a controlled environment for isolating how LLMs learn from reflection, which would be difficult to assess in more diverse clinical corpora. The LLMs were tasked with extracting the clinical variables listed in Table I.

All clinical annotations in this dataset were performed by clinical staff under the supervision of a Specialist in Neurology. The notes have been fully deidentified to ensure compliance with privacy and ethical standards. The data collection protocol was reviewed and approved by the institutional research ethics committee, approval number YX200651. Written informed consent was obtained from all participants prior to their inclusion in the study.

The basic prompt used in our study as baseline included variable definitions, task description, two-shot examples, clinical note, and an output template. The LLMs were prompted to follow a chain-of-thought approach. The two-shot examples were generated using ChatGPT-4o and contain both the correct answers and reasoning steps, which were validated by clinical staff. Example of the standard prompt is shown in textbox 1.

### B. Reflection framework

Many widely used medical benchmarks, such as MedQA, span broad medical domains where insights from one case rarely generalizable to another. In contrast, our task focuses on extracting a fixed set of variables from clinical notes that share structural and contextual similarities. Because of this consistency, extraction errors tend to follow recurring

---

**Textbox 1: Standard prompt example**

For First pass method:
You are provided with clinical notes detailing an Endovascular Thrombectomy procedure. Your task is to extract the **"first pass method"** used for thrombus removal.
The possible methods are:
- Stentriever
- Aspiration
- Solumbra (Stentriever together with Aspiration without indication of failure in between)
Based on your analysis, specify the first pass method as either Aspiration, Stentriever, or Solumbra.
Output your final answer in this form **Final answer: [[Aspiration or Stentriever or Solumbra]]**
Example 1: [Example 1 insert]
Example 2: [Example 2 insert]
Now, analyze the following clinical notes and determine the "first pass method" used for thrombus removal, following the instructions above.
Clinical Note: [clinical note insert]

---

patterns, enabling systematic improvement through reusable insights. This assumption underpins our method: instead of treating each reflection as a standalone process, we consolidate multiple reflection outputs into an optimized supporting statement that can guide future extractions more effectively. This process is conducted separately for each variable, allowing optimizations to address variable-specific error patterns.

In this study, a case refers to the process of extracting a single variable from a clinical note. In our method, self-reflection is applied to incorrect extractions from baseline prompts. To guide the reflection process, three distinct types of information were provided separately: (1) **Ground Truth Reflection (GTR)**: Only the annotated ground truth answer was provided for the reflection. This approach is the least resource-intensive, as the answers are already available in the annotations. However, it introduces a risk of answer leakage, where the model may produce the right answer simply due to exposure, rather than through a meaningful reflection on its reasoning. (2) **Chain-of-Thought with Answer (CoTA)**: the model is shown an output from another LLM that includes both a reasoning path and the correct answer. This method aims to improve reflection quality by exposing the model to accurate reasoning patterns. (3) **Chain-of-Thought Only (CoTO)**: Similar to CoTA but with the final correct answer masked. This aims to minimize the risk of answer leakage while still leveraging the reasoning path to enhance the reflection process.

The reflection process is repeated up to five times or until the correct answer is obtained. This limit was set based on preliminary observations that showed diminishing returns and a higher likelihood of repetitive, uninformative reflections beyond five attempts. We use a temperature of 0.0 for extraction to ensure deterministic output, and 0.8 for self-reflection to encourage diverse, creative suggestions. This setup increases the likelihood of generating alternative insights when earlier attempts fail. The prompt templates used for reflection and consolidation are shown in textbox 2, 3.

These reflection outputs are summarized and consolidated into an optimized supporting statement to improve future

TABLE I
DESCRIPTION OF THE SEVEN CLINICAL VARIABLES EXTRACTED FROM ENDOVASCULAR THROMBECTOMY NOTES.

| Variable | Description | Answer Type |
|---|---|---|
| Anesthesia | Type of anesthesia used | Local, General |
| First pass method | Technique used during the first attempt to remove the clot | Stentriever, Aspiration, Solumbra |
| Number of passes | Total attempts made to retrieve the clot | Numerical |
| Rescue method | Backup technique used if standard clot removal fails | N/A, Stentriever, Aspiration, Solumbra |
| Tandem lesions | Whether the patient had both extracranial and intracranial arterial blockages | Yes, No |
| Angioplasty | Whether balloon dilation was used to treat a vessel during the procedure | Yes, No |
| mTICI | Score measuring blood flow restoration after thrombectomy | 0, 1, 2a, 2b, 2c, 3 |

**Textbox 2: Reflection Prompt**

This is attempt # at improving the original prompt.
Your task is to:
1. Re-examine the original prompt in light of the previous reflections and the latest incorrect response.
2. Propose new, more creative improvements or refinements to the prompt that have not been suggested before.
3. Provide concise, actionable suggestions in point form for refining the prompt—avoid suggesting a fully rewritten prompt.
Example Output template:
<<Explanation:>>
Supporting statements:
• Here is a clarification of concept "A" ...
Original prompt:
`[original prompt insert]`
Iteration number: `[iteration number insert]`
Previous reflections:
`[previous reflections insert]`
Latest incorrect response:
`[incorrect response insert]`
Correct answer:
`[correct answer insert]`

**Textbox 3: Consolidation Prompt**

You have been given several reflection outputs, each analyzing errors made by an LLM in response to a particular prompt. Your task is to:
1. Review all of the provided reflection outputs.
2. Identify and list the distinct types of errors noted across the reflections.
3. Summarize these errors in a clear, point-form format. Be concise in your response; however, capture all of the essential information.
4. Where available, include any provided "supporting statements" that guide the LLM toward correcting these errors. Integrate or reference these supporting statements in a cohesive manner.
Then separately provide a refined summary that can be added to the original prompt to improve the overall performance after the tag below.
`[refined Summary]`
Reflection outputs:
`[reflection outputs insert]`

The workflow for integrating self-reflection and consolidation is illustrated in Fig. 1. The key steps are as follows:

1) The standard prompt utilizes chain-of-thought reasoning and two-shot examples with corresponding intermediate reasoning steps.
2) The standard prompt is applied to each clinical note individually.
3) Cases in which the LLM failed to extract the correct answer were flagged for self-reflection. We tested three types of guidance: (1) annotated ground truth (**GTR**), (2) a correct chain-of-thought and final answer generated by another LLM (**CoTA**), and (3) CoTA with the final answer masked (**CoTO**).
4) Using the provided guidance, the LLM generates a reflection.
5) The reflection is inserted into the original prompt and used to retry the same case.
6) If the reflection does not yield the correct answer, it is retained to inform subsequent reflection attempts. Up to five attempts are made, with the goal of avoiding repeated errors and encouraging alternative reasoning paths.
7) Only the final reflection that leads to a correct extraction is kept for consolidation.
8) For each variable, up to 10 successful reflections are consolidated into a single supporting statement, which is added to the original prompt. This enhanced prompt is then re-applied to all clinical notes to assess its effect on extraction performance.

While commercial models often achieve state-of-the-art results [30], their use in healthcare faces significant practical barriers. Patient data privacy regulations often prohibit transmitting clinical notes to external APIs, and per-token pricing models can become prohibitive for large-scale extraction tasks. Open-source LLMs, particularly those in the 7B-72B parameter range, support local deployment, allow more control, and can be fine-tuned for specific domains. For our experiments, we used models from Llama-3 (8B, 70B) [31], Gemma-2 (9B, 27B) [32], and Qwen2.5 (7B, 72B) [33]. Although these models may exhibit lower baseline performance compared to their commercial counterparts, they represent the realistic deployment scenario. Our study therefore focuses on these practically deployable models, as understanding their limitations and optimization potential is crucial for real-world clinical applications.

extractions. Due to context length constraints (e.g., Gemma2's 8k token limit), we cap the consolidation set at 10 reflections, a number determined to balance the need for diverse insights against the context length limitations of the tested models. This is a practical necessity: certain variables can produce up to 50 incorrect cases, with each reflection spanning several hundred tokens. Without summarization, the input would soon exceed the context window. The key motivation for leveraging LLM's summarization ability is that across $n$ reflections, there may be anywhere from 1 to $n$ distinct error types, some recurring, others entirely unique. Summarization is therefore essential for identifying and prioritizing the most informative error patterns, ensuring that the consolidated statement provides targeted guidance for future extractions.

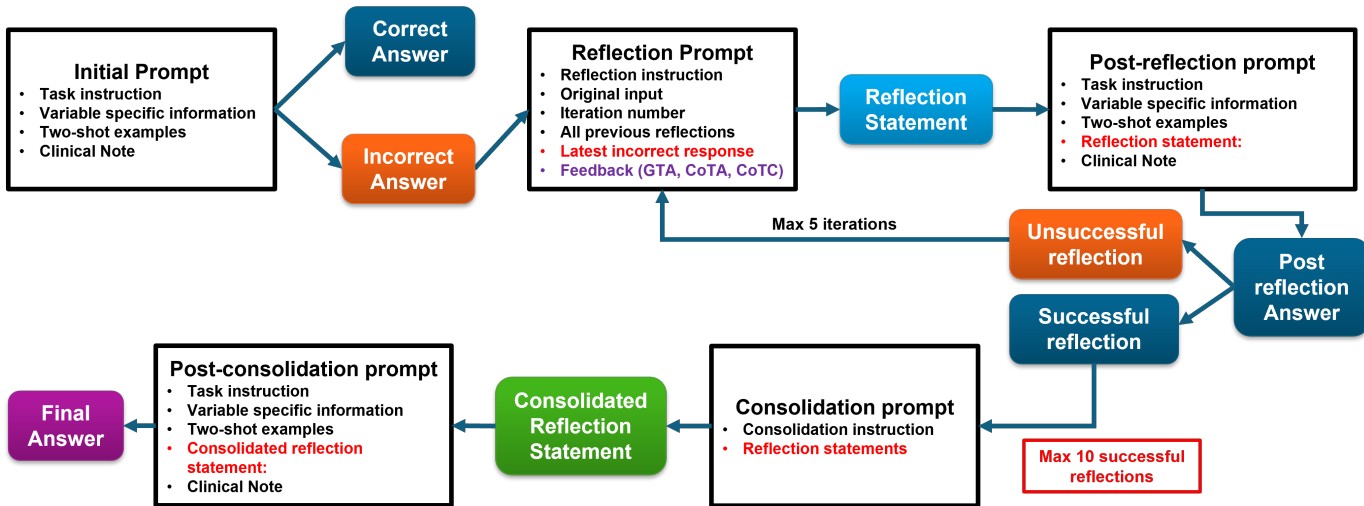

Fig. 1. Overview of the self-reflection framework for clinical information extraction. Three reflection guidance types (GTR, CoTA, CoTO) are tested to evaluate their effect on LLM performance. The process allows up to five iterations, with successful reflections consolidated to guide future extractions.

## III. RESULT

### A. Baseline performance

The baseline performance of the LLMs on each variable is shown in Table II. As expected, larger models within the same foundation model family tend to perform better. Qwen2.5-72B outperformed the other models overall. The models show strong performance on variables such as anesthesia, angioplasty, and mTICI, with F1 scores ranging from 83.1 to 97.4. In contrast, variables related to the thrombus removal process, such as "first pass method", "number of passes", and "rescue method" require more complex reasoning. For example, if both a stentriever and aspiration are used during a procedure, the model must determine whether they were applied sequentially (counting as two separate passes) or in combination (counted as a single Solumbra attempt). Performance on these variables shows greater variation, with accuracy ranging from 0.3 to 0.9.

### B. Effect of reflection

Figure 2 presents the performance of each model in correcting extraction errors after one and five iterations of self-reflection (up to step 6 of the workflow). All models benefit from self-reflection, with a clear increase in resolved cases after five rounds. For example, Llama-3-8B improves from 49.1% to 70.0%, a 20.9% gain. In contrast, Qwen2.5-72B sees a smaller rise from 54.6% to 63.0%, likely due to its already high baseline accuracy.

A consistent trend emerges across all six models: the GTR approach yields the highest correction rates, followed by CoTA, with CoTO performing worst. This pattern suggests that access to the final correct answer during reflection, as in GTR and CoTA, enhances correction effectiveness, though potentially at the cost of answer leakage. The weaker performance of CoTO indicates that while reasoning paths are valuable, the absence of the answer may limit their utility in guiding accurate reflection.

Next, we evaluate how consolidating multiple reflections into a generalized prompt affects overall performance. The goal here is not to improve a single case, but to derive transferable insights that improve extraction performance across all cases for the same variable. Figure 3 presents the overall performance of the proposed reflection algorithm across different models and guidance strategies.

As shown in the previous section, larger models generally yield better performance across all three foundational architectures. However, none of the three reflection approaches (GTR, CoTA, and CoTO) consistently outperform the baseline. Some modest improvements are observed, particularly with the GTR approach. For example, Llama-3-8B's F1 score rises from 66.9% to 72.6%. Similarly, Qwen2.5-7B improves its baseline F1 from 76.9% to 79.6%. While high-performing models like Llama-3-70B and Qwen2.5-72B show little to no improvement or even slight regressions. This suggests that the process of LLM-driven consolidation, as implemented, may introduce its own errors or dilute the specific insights that were effective at the individual reflection level. Moreover, there is no clear ranking among the three reflection strategies. Although GTR achieves the highest performance gains in some models, it does not consistently surpass CoTA or CoTO.

Our results on individual reflections suggest that LLMs can correct their own errors through self-reflection. However, the insights gained at this level are not well maintained during consolidation. This reveals a fundamental limitation in LLMs' ability to generalize subtle clinical reasoning across cases, suggesting that reflection mechanisms may require domain-specific adaptation rather than generic approaches. These limitations ultimately constrain the utility of reflection-based optimization in specialized clinical contexts.

To assess the impact of consolidation, we compared differences in F1 score across seven extracted variables between the three reflection approaches and the baseline (Figure 4). For most variables, especially those with high baseline accu-

| Macro F1 Score | Llama-3-8B | Llama-3-70B | Gemma-2-9B | Gemma-2-27B | Qwen2.5-7B | Qwen2.5-72B |
|---|---|---|---|---|---|---|
| Anesthesia | 83.1 | 94.1 | 90.4 | 89.6 | 93.3 | 94.1 |
| First pass method | 31.8 | 75.6 | 71.2 | 77.6 | 64.0 | 88.0 |
| Number of passes | 57.7 | 85.6 | 85.4 | 85.5 | 79.7 | 90.4 |
| Rescue method | 37.9 | 51.7 | 37.4 | 37.3 | 50.5 | 78.2 |
| Tandem lesions | 75.7 | 78.0 | 73.1 | 69.5 | 70.4 | 73.6 |
| Angioplasty | 95.2 | 97.4 | 95.6 | 94.8 | 93.9 | 93.0 |
| mTICI | 86.6 | 84.8 | 86.4 | 85.2 | 86.4 | 85.2 |

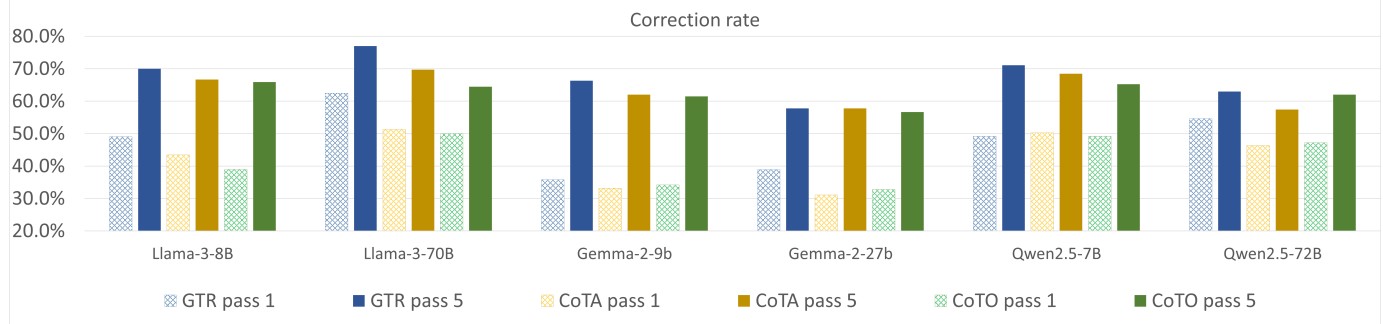

Fig. 2. Percentage of incorrect extractions successfully corrected after one and five iterations of self-reflection across different models and reflection approaches (GTR, CoTA, CoTO).

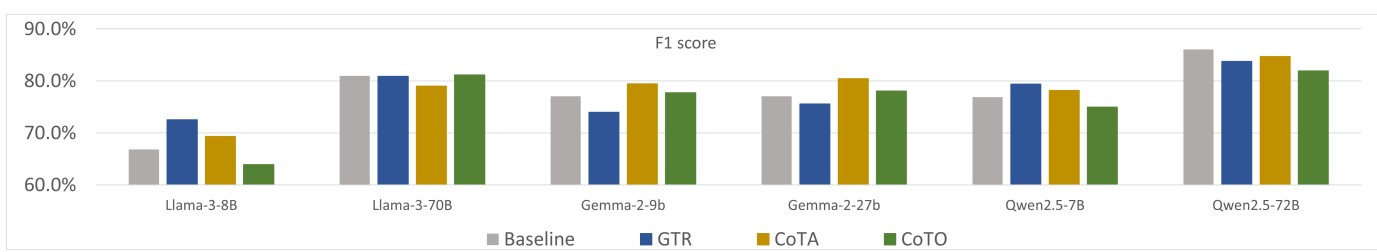

Fig. 3. Comparison of overall model performance (F1 score) across baseline prompting and three reflection approaches (GTR, CoTA, CoTO).

racy like 'mTICI' and 'Angioplasty', F1 differences remained within a few percentage points, indicating the LLMs' robustness in extracting these terms. In some cases, the difference was exactly 0%, which corresponds to the models failed to correct any of the original errors in the individual reflection stage. Notably, within the same model, all three reflection approaches tended to show consistent effects, either all positive or all negative, likely because they shared a similar set of cases for consolidation.

To understand why consolidation sometimes degrades performance, we examined the "Anesthesia" variable as an example. This variable is relatively simple compared to others, with a baseline accuracy of roughly 90% across models. Under the GTR approach, Gemma2-9b's performance declined post-consolidation, with 25 out of 30 new errors misclassifying "local anesthesia" as "general anesthesia." During individual reflection, when the correct annotation was provided, the LLM correctly identified that "propofol in combination with dexmedetomidine used for intravenous basal anesthesia" describes a form of sedation rather than general anesthesia. When this reflection statement was incorporated into the standard prompt, the LLM successfully produced the correct answer. However, in the consolidation, this insight was reduced to: "The prompt should clearly define sedation and general anesthesia or provide additional context to prevent misinterpretations." This demonstrates a clear instance where the LLM's summarization for consolidation led to an over-generalization, losing the specific, actionable detail from the successful individual reflection. Once this more generic guidance was applied across the dataset, the model failed to consistently distinguish between sedation and general anesthesia in similar notes. It repeatedly defaulted to "general anesthesia" whenever sedatives were mentioned together with local anesthesia. This example illustrates the challenge of encoding fine-grained domain knowledge during prompt consolidation and reinforces the need for expert oversight to prevent loss of essential details during optimization.

While consolidation sometimes degraded performance, it could also lead to significant improvements. A notable success was observed for the variable first pass method using Llama-3-8B. In the baseline setting, the model correctly identified only 35 out of 115 cases, with a particularly poor performance on Solumbra, correctly classified in only 17 out of 74 instances. After applying GTR-based reflection consolidation, accuracy

| Model | Variable | GTR | CoTA | CoTO |
|---|---|---|---|---|
| Llama-3-8B | Anesthesia | 5.9% | -1.6% | 1.5% |
| | First pass method | 36.3% | 35.1% | 27.1% |
| | Number of passes | -10.4% | -8.7% | 2.6% |
| | Rescue method | 9.8% | 6.9% | 11.2% |
| | Tandem lesions | 1.7% | -7.8% | -25.9% |
| | Angioplasty | -0.4% | -5.6% | -36.5% |
| | mTICI | -2.8% | -1.1% | -0.3% |
| Llama-3-70B | Anesthesia | -0.6% | -1.0% | -1.0% |
| | First pass method | -13.6% | 3.7% | 3.9% |
| | Number of passes | 2.7% | -4.6% | -4.2% |
| | Rescue method | 8.7% | -5.3% | 6.0% |
| | Tandem lesions | 0.4% | -6.6% | -5.1% |
| | Angioplasty | 0.9% | 0.0% | 0.9% |
| | mTICI | 0.8% | 0.0% | 0.7% |
| Gemma-2-9b | Anesthesia | -14.7% | 0.8% | -10.1% |
| | First pass method | -6.6% | 4.6% | 5.9% |
| | Number of passes | -1.3% | -1.4% | -3.6% |
| | Rescue method | 4.3% | 12.9% | 9.5% |
| | Tandem lesions | -2.8% | -0.5% | 0.6% |
| | Angioplasty | -0.1% | 0.9% | 0.9% |
| | mTICI | -0.5% | -0.8% | 1.7% |
| Gemma-2-27b | Anesthesia | 3.5% | 4.5% | 3.9% |
| | First pass method | -15.6% | 1.3% | -0.5% |
| | Number of passes | -8.4% | -5.7% | -1.1% |
| | Rescue method | 22.2% | 21.6% | 2.3% |
| | Tandem lesions | -12.2% | -0.7% | -0.1% |
| | Angioplasty | 0.9% | 2.6% | 0.0% |
| | mTICI | -0.6% | 0.5% | 2.5% |
| Qwen2.5-7B | Anesthesia | -2.3% | -4.3% | -0.2% |
| | First pass method | 9.0% | 8.4% | 3.6% |
| | Number of passes | 3.0% | 3.0% | -8.7% |
| | Rescue method | 6.0% | -0.2% | -4.1% |
| | Tandem lesions | 1.7% | 1.4% | -1.1% |
| | Angioplasty | 1.8% | 1.8% | -1.7% |
| | mTICI | -1.3% | -0.8% | -1.3% |
| Qwen2.5-72B | Anesthesia | 0.0% | 0.0% | 0.0% |
| | First pass method | -2.0% | -3.0% | -1.7% |
| | Number of passes | -17.7% | -3.1% | -14.9% |
| | Rescue method | -5.0% | -16.4% | -18.2% |
| | Tandem lesions | 2.4% | 6.8% | 3.2% |
| | Angioplasty | 3.6% | 5.3% | 4.4% |
| | mTICI | 2.6% | 0.9% | -1.8% |

Fig. 4. Variable-level F1 score differences after applying three reflection consolidation strategies compared to the baseline prompt.

on Solumbra cases rose to 71 out of 74, illustrating the potential of targeted reflection generalization when it aligns well with model weaknesses.

## IV. DISCUSSION

As healthcare increasingly depends on structured data extraction from clinical documentation, our study highlights key challenges in building self-improving extraction systems. To explore this, we introduce a domain-specific dataset of 115 endovascular thrombectomy notes and evaluate reflection-based prompting strategies tailored to specialized clinical concepts. Our results demonstrate that LLMs can identify and correct errors through self-reflection when given sufficient information. However, we encountered significant limitations in the consolidation stage, where reflections were summarized into a generalized statement. This process frequently diluted key clinical insights, diminishing the effectiveness of reflection generalization. To address this issue, integrating a human-in-the-loop (HITL) [34] could help retain clinically relevant information. While the necessity of HITL oversight in medical AI is well-established, our main contribution is in empirically identifying where current LLM-based self-correction and generalization fail in clinical information extraction settings. This finding pinpoints consolidation as a critical workflow stage where targeted human oversight remains essential, providing actionable insights for designing future clinical NLP systems.

A practical HITL extension to our framework would introduce an expert review step following the eight-point reflection process described in the Methods section. Specifically, after the LLM consolidates successful reflections into a supporting statement for each variable (Step 8), a human expert would review this consolidated output to ensure preservation of key clinical details and reasoning. The reviewer may edit or annotate statements as needed before incorporation into enhanced prompts. For example, if a consolidation ambiguously distinguishes between "sedation" and "general anesthesia," the reviewer could refine the statement to provide explicit definitions and clarify clinically relevant scenarios. As this oversight targets only a small number of consolidated statements per variable, the added human effort remains modest. Such hybrid pipelines promise better alignment with clinical standards, particularly in specialized domains like thrombectomy, where accurate generalization requires both contextual understanding and domain sensitivity.

In terms of computation cost, for each variable, we allowed up to 10 successful reflections, each taking up to 5 attempts (reflection generation and validation). This results in up to 100 reflection-related queries per variable, plus one consolidation query, about 101 queries per variable during consolidation. After this, inference on new cases only needs adding the consolidated supporting statement to the prompt, so the per-case cost is only slightly higher. In our approach, most of the extra token usage happens only during the initial consolidation phase. However, the gains from reflection-based consolidation were modest and inconsistent in our results, so the cost-benefit tradeoff is not clearly favorable. Further work is needed to improve efficiency and effectiveness before these pipelines can be widely adopted in clinical practice.

Overall, our findings suggest that self-reflection alone is insufficient for reliable knowledge distillation in high-stakes clinical domains. Human-in-the-loop oversight is not only a safety mechanism but an essential component for translating LLM reasoning into trustworthy clinical tools. Future research could also explore alternative mechanisms for consolidation beyond LLM-driven summarization, such as structured knowledge extraction from reflections or template-based synthesis guided by human experts, to better preserve critical details.

We chose summarization as our consolidation method because it is intuitive, widely used, and requires minimal annotation, providing a straightforward baseline to evaluate an LLM's generalization capability. More sophisticated methods, such as rule induction, error clustering, or dynamic example creation, might offer deeper insights but require detailed intermediate annotations (e.g., explicit reasoning chains). Our

current dataset only include final answers, which limits our ability to develop and systematically evaluate these advanced strategies. Future studies should incorporate detailed annotations to enable the exploration and comparison of alternative consolidation techniques.

We acknowledge several methodological limitations. First, due to the labor-intensive nature of manually reviewing reflections and the lack of intermediate annotations in our dataset, we could not directly evaluate and select truly informative reflections. Consequently, reflections were deemed "successful" solely based on achieving correct final extractions. Therefore, some "successful" reflections may be minimally informative.

Second, we were similarly unable to explicitly evaluate the risk of answer leakage. Resource limitations prevented extensive manual review across numerous reflections, forcing reliance on indirect assessments through the final accuracy. These results cannot fully isolate answer leakage from other factors, such as generalization limitations. Future research should incorporate explicit qualitative analysis focusing on answer leakage.

Third, our two-shot examples were generated using a high-performance proprietary model (ChatGPT-4o). While validated by clinicians, this approach may introduce a stylistic bias in the prompts, and the performance of the tested open-source models could potentially differ if the examples were generated by the models themselves.

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
