# OpenReview forum: "Limitations of Automated Reflection Consolidation in LLMs for Clinical Note Extraction: Evidence for Human-in-the-Loop Requirements"
_IEEE.org/EMBS/BHI/2025/Conference — BHI 2025_

### Official Review · Reviewer_F64h · 2025-07-02
**Limitations of Automated Reflection Consolidation in LLMs for Clinical Note Extraction: Evidence for Human-in-the-Loop Requirements**

**Confidence:** 3
**Clarity Of Writing:** good
**Clinical Significance:** good
**Methodological Novelty:** fair
**Overall Rating:** 5

**Experiments And Results:**

good

**Questions For The Authors:**

Add more methodological detail on how reflections were evaluated, how consolidation statements were curated, and how reflection components were judged as successful or not.

Quantify or at least estimate the token and compute cost of the reflection pipeline, and discuss whether the gains justify the cost at scale.

Consider revising the Discussion opening to lead with the paper’s strongest finding before addressing limitations.

Explore visual enhancements (e.g., hybrid heatmap + table) to better highlight model performance patterns.

Provide at least a conceptual framework or example of how human-in-the-loop oversight might work in practice and what its cost/effort profile might look like.

**Strengths:**

The paper addresses an important problem in clinical NLP: improving structured data extraction from complex, unstructured notes.

It takes on a challenging and high-stakes domain (endovascular thrombectomy) where accurate information extraction could have real clinical value.

The use of a homogeneous dataset is well-justified and allows for controlled analysis.

The paper provides a thoughtful discussion of the limitations of reflection consolidation and makes a reasoned call for human-in-the-loop integration.

The authors’ transparency about where their method failed is commendable.

**Summary Of The Paper:**

The paper explores whether reflection-based prompting strategies can improve structured information extraction from clinical notes describing endovascular thrombectomy procedures. The authors evaluate three reflection strategies (Ground Truth Reflection, Chain-of-Thought + Answer, Chain-of-Thought Only) using several open-source LLMs (e.g., LLaMA-3, Qwen2.5, Gemma-2) and assess performance on seven key clinical variables. Results show that while reflection can improve extraction in isolated cases, attempts to consolidate reflections into generalizable prompt components often dilute useful detail and lead to mixed or degraded performance. The authors argue that human-in-the-loop oversight is needed for optimal results.

**Weaknesses:**

The methods section lacks detail on how reflections were selected, consolidated, and judged for success, limiting reproducibility.

The performance improvements from reflection are modest and inconsistent, with consolidation often leading to worse results.

The paper does not address the token cost or compute cost of their approach, nor does it weigh this cost against the observed gains. Given the token-rich nature of reflection-consolidation, applying this at hospital scale would likely be financially prohibitive without clear demonstrated value.

No concrete plan is offered for operationalizing the human-in-the-loop component, nor is the potential human effort cost quantified.

The Discussion starts by emphasizing limitations rather than opening with the key positive finding, which weakens the paper’s framing.

A clearer, more informative visualization (e.g., hybrid heatmap + table for Table 2) would have helped readers quickly grasp the pattern of model performance.

---

### Official Review · Reviewer_JFHi · 2025-07-09
**This paper provides a valuable, empirically-grounded argument for the necessity of human oversight in clinical AI systems. By demonstrating that an LLM's attempt to autonomously consolidate its own learnings can fail, the authors make a significant and timely contribution. While the study design has certain limitations that open up exciting avenues for future work, its core finding is well-supported and represents a solid step forward for the field.**

**Confidence:** 4
**Clarity Of Writing:** fair
**Clinical Significance:** good
**Methodological Novelty:** good
**Overall Rating:** 7

**Experiments And Results:**

good

**Questions For The Authors:**

To help readers fully appreciate the context of your findings, it would be good to elaborate on the choice of summarization as the initial method for consolidation. Adding this rationale would further strengthen the discussion of the paper.

**Strengths:**

The primary strength of the work is its focus on a novel and critical research question: Can LLMs generalize from their own mistakes to create a durably better system? The study is well designed, using a homogeneous dataset that provides a realistic testbed for this hypothesis. Furthermore, the paper excels by not just reporting failure but clearly diagnosing its cause, showing how the LLM's summarization process discards essential clinical details.

**Summary Of The Paper:**

The authors investigate whether LLMs can autonomously improve clinical information extraction. Their framework has LLMs identify and correct their own errors on a dataset of 115 endovascular thrombectomy notes. Successful corrections, called reflections, are then automatically summarized by the LLM into a generalized statement intended to improve future performance. The authors find that while this reflection process helps correct individual errors, the consolidated summaries often lose essential clinical nuance, leading to inconsistent and sometimes worse overall performance. They conclude that fully autonomous generalization is currently unreliable and requires human oversight.

**Weaknesses:**

The paper provides a strong foundation that invites interesting future research directions. The authors choice to focus on summarization as the consolidation method establishes an important baseline. A natural next step for the field would be to explore how this approach compares to more structured consolidation methods, such as rule induction or dynamic example creation. This could clarify whether the challenge lies in summarization specifically or in automation more broadly. Similarly, the current study provides a strong benchmark on a focused dataset; future work could build upon these findings by applying the framework to larger corpora to investigate how generalization performance scales with more data.

---

### Official Review · Reviewer_iexg · 2025-07-13
**A Valuable but Methodologically Limited Study on the Generalization Failures of LLM Self-Correction in a Clinical Setting**

**Confidence:** 4
**Clarity Of Writing:** fair
**Clinical Significance:** good
**Methodological Novelty:** poor
**Overall Rating:** 5
**Final Rating:** 5

**Experiments And Results:**

good

**Questions For The Authors:**

The study's final core conclusion is that "in high-risk clinical domains, a HITL is required". In the field of medical AI, this is almost a recognized truism, not a novel discovery. A large body of existing literature has already emphasized the necessity of human supervision in automated systems, especially LLMs. Its marginal knowledge contribution very limited.

**Strengths:**

The central contribution is the clear empirical distinction between an LLM's ability to self-correct a single instance versus its ability to generalize that correction.

**Summary Of The Paper:**

This paper investigates the ability of open-source LLMs to autonomously improve their performance on a specialized clinical information extraction task through a process of self-reflection and generalization. The authors use a dataset of 115 clinical notes from endovascular thrombectomy procedures, tasking six different LLMs with extracting seven key variables.

**Weaknesses:**

The study finds that while all models show improved performance in correcting individual errors through reflection, the consolidation of these reflections leads to mixed and inconsistent results for overall performance. Many models show only modest gains or even performance degradation. The paper acknowledges the risk of "answer leakage" but does not sufficiently address how this might confound the results.